# Coagulation System Activation for Targeting of COVID-19: Insights into Anticoagulants, Vaccine-Loaded Nanoparticles, and Hypercoagulability in COVID-19 Vaccines

**DOI:** 10.3390/v14020228

**Published:** 2022-01-24

**Authors:** Mohamed S. Abdel-Bakky, Elham Amin, Mohamed G. Ewees, Nesreen I. Mahmoud, Hamdoon A. Mohammed, Waleed M. Altowayan, Ahmed A. H. Abdellatif

**Affiliations:** 1Department of Pharmacology and Toxicology, College of Pharmacy, Qassim University, Qassim 52471, Saudi Arabia; m.abdelbakky@qu.edu.sa; 2Department of Pharmacology and Toxicology, Faculty of Pharmacy, Al-Azhar University, Cairo 11884, Egypt; 3Department of Pharmacognosy, Faculty of Pharmacy, Beni-Suef University, Beni-Suef 62514, Egypt; El.Saleh@qu.edu.sa; 4Department of Medicinal Chemistry and Pharmacognosy, College of Pharmacy, Qassim University, Qassim 52471, Saudi Arabia; ham.mohammed@qu.edu.sa; 5Department of Pharmacology and Toxicology, Faculty of Pharmacy, Nahda University, Beni-Suef 11787, Egypt; mohamed_pharma86@yahoo.com (M.G.E.); nesreen.mahmoud@nub.edu.eg (N.I.M.); 6Department of Pharmacognosy, Faculty of Pharmacy, Al-Azhar University, Cairo 11884, Egypt; 7Department of Pharmacy Practice, College of Pharmacy, Qassim University, Qassim 52471, Saudi Arabia; w.altowayan@qu.edu.sa; 8Department of Pharmaceutics, College of Pharmacy, Qassim University, Qasssim 52471, Saudi Arabia; 9Department of Pharmaceutics and Pharmaceutical Technology, Faculty of Pharmacy, Al-Azhar University, Assiut 71524, Egypt

**Keywords:** hypercoagulability, COVID-19, natural anticoagulants, nanoparticles, targeting, COVID-19 vaccines

## Abstract

The severe acute respiratory syndrome coronavirus 2 (SARS-CoV-2), also known as COVID-19, is currently developing into a rapidly disseminating and an overwhelming worldwide pandemic. In severe COVID-19 cases, hypercoagulability and inflammation are two crucial complications responsible for poor prognosis and mortality. In addition, coagulation system activation and inflammation overlap and produce life-threatening complications, including coagulopathy and cytokine storm, which are associated with overproduction of cytokines and activation of the immune system; they might be a lead cause of organ damage. However, patients with severe COVID-19 who received anticoagulant therapy had lower mortality, especially with elevated D-dimer or fibrin degradation products (FDP). In this regard, the discovery of natural products with anticoagulant potential may help mitigate the numerous side effects of the available synthetic drugs. This review sheds light on blood coagulation and its impact on the complication associated with COVID-19. Furthermore, the sources of natural anticoagulants, the role of nanoparticle formulation in this outbreak, and the prevalence of thrombosis with thrombocytopenia syndrome (TTS) after COVID-19 vaccines are also reviewed. These combined data provide many research ideas related to the possibility of using these anticoagulant agents as a treatment to relieve acute symptoms of COVID-19 infection.

## 1. Introduction

The most common symptoms observed in COVID-19 patients are fever, fatigue, and cough, which are mostly associated with less frequent symptoms such as headache, dyspnea, skin rashes, sore throat, diarrhea, anosmia, and nausea [1]. Around 80% of COVID-19 patients do not need hospitalization [2]. Owing to COVID-19 infection, severe acute respiratory syndrome coronavirus (SARS-CoV) binds to angiotensin-converting enzyme 2 receptor (ACE2), which is widely expressed all over the body, such as arterial smooth muscle cells in many organs, such as lung type II alveolar cells, enterocytes of the small intestine, arterial and venous endothelial cells, the neural cortex, and the brainstem [3,4]. Interestingly, several systems, such as coagulation [5,6] and immune systems [6], are activated after COVID-19 infection. However, the correlation between COVID-19 infection and activation of the coagulation system still requires elucidation. RNA virus infections, such as COVID-19, flavivirus (Dengue fever), filovirus (Ebola), and arenavirus (Lassa) [7], are reported to activate the coagulation system. Therefore, previous data might help understand the correlation between hypercoagulability and COVID-19.

Thrombocytopenia and increased blood levels of D-dimer have been reported in 36.2% and 46.4% of the total reported patients, respectively, and become even higher in severe cases [1]. Although earlier research supported the idea that disseminated intravascular coagulation (DIC) was connected to the increased mortality in COVID-19 pneumonia [8], DIC is generally uncommon in COVID-19 patients [9]. Moreover, the percentage of complicated COVID-19 cases by DIC has been found to be 0.6% for survivors and 71.4% for non-survivors [10]. In addition, prognosis outcome is directly correlated with prothrombin time (PT) and the levels of fibrin degradation products (FDP) in COVID-19 infected patients [10]. Moreover, decreased platelet counts (thrombocytopenia) have also been associated with severe COVID-19 [11]. Activation and elevation of the immune system, Toll-like receptors, Willebrand factor, and endothelial dysfunction are all involved in the pathogenesis of viral infection [12,13,14,15]. Platelets’ interaction, white blood cells (WBCs), and endothelial cells play a pivotal role in the activation of the coagulation system in different viral infections [16,17]. Nanotechnology is considered a viable medicine delivery method. A variety of biological applications in vitro and in vivo have been shown to be suitable for their bioactive qualities [2]. The intense interest in nanoparticles and their derivatives has increased therapy options. Applications of nanotechnology in medicine, biomedical research, and biotechnology include cancer cell-targeted therapy and other biomedical treatments [18]. Furthermore, nanotechnology can help prevent COVID-19 by employing vaccine-loaded nanoparticles [19]. However, the nanoparticles have effect on the blood coagulation system have received little consideration. Nanoparticles that interact with the blood coagulation system can have both positive and negative effects on the host [20]. The relationship between COVID-19 and hypercoagulability, specific hypercoagulability markers, anticoagulant therapeutic approaches, natural anticoagulants, and nanoparticle formulations of COVID-19 drugs such as anticoagulants are reviewed in this article.

### 1.1. Coagulation System Activation in Different Diseases

Several investigations have focused on the proinflammatory role of the coagulation system that leads to non-hematological pathological conditions (Figure 1). For example, Lorenzano et al., (2019) reported that there is a strong correlation between the deposition of different coagulation factors in the CNS and neuro-inflammation and neurodegeneration [21]. In addition, Davalos et al., (2019) reported that fibrinogen passes the blood–brain barrier (BBB) when it is disrupted and consequently spreads and converts into fibrin in the CNS; this triggers an inflammatory response and immune activation [22]. Moreover, this cytotoxicity has been observed in autoimmune encephalomyelitis and multiple sclerosis (MS) pathogenesis [23].

Interestingly, cancer patients are at high risk of endothelial injury and an imbalance between pro- and anti-thrombotic factors, leading to a hypercoaguable state. This may be caused by patient-specific, chemotherapy-related, and tumor-specific factors resulting in the deposition of intravascular fibrin with depletion of blood clotting factors and development of thrombocytopenia [24]. Tissue factor (TF) is also overexpressed in different types of tumors such as colorectal, gastric, pancreatic, and lung cancer [25]. It has been demonstrated by Abdel-Bakky et al., (2011) that TF mediates liver toxicity in the monocrotaline/lipopolysaccharide mice model [26]. Similarly, TF blocking by deoxyoligonuceotides antisense is successfully prevented by thioacetamide [27], CCL4-induced liver injury in mice [28], and monocrotaline/lipopolysaccharide-induced renal toxicity in mice [29].

Therefore, targeting coagulation systems is an important potential therapeutic tool for a wide range of diseases. The antiplatelet drug clopidogrel and/or the thrombin inhibitor dabigatran can be used as a prophylactic approach against liver fibrosis [30]. Ewees et al., (2018) suggested that the use of the anticoagulant, rivaroxaban, factor Xa (FXa) inhibitor could be a useful protection against cisplatin-induced acute tubular necrosis [31]. Dabigatran, the direct thrombin inhibitor, has also shown efficacy in a breast cancer murine model [32]. Side by side, FXa inhibitors are used in clinical trials to decrease coagulation cascade activation and to increase microvascular circulation in sickle cell disease (SCD) [33].

Moreover, the “Middle East respiratory syndrome” (MERS-CoV) was initially identified in Saudi Arabia in 2012 and was associated with thrombotic complications and hematologic manifestations [5]. Thrombocytopenia was reported in 36% of 47 laboratory-confirmed MERS-CoV cases in Saudi Arabia [34]. Interestingly, clinical and post-mortem examination reports of COVID-19 patients from China and the United States have revealed increased clotting and scattered intravascular coagulation with small vessel thrombosis and pulmonary infarction [35,36]. Furthermore, low platelet levels, elevated D-dimer, and prolonged PT relate to worse results in patients with COVID-19 [10].

Viral infection triggers an inflammatory immune response, causing the activation of the coagulation system [5]. The excessive release of coagulation factors is controlled by negative feedback and physiological anticoagulants such as TF inhibitor, anti-thrombin, and the protein C system. Increased consumption of physiological anticoagulants disrupts procoagulant and anticoagulant homeostatic mechanisms, resulting in D-dimer elevation and the development of micro thrombosis with scattered intravascular coagulation in severe COVID-19 [37,38]. Therefore, we suggest that it is a good practice to use anticoagulant therapy carefully in calculated and controlled doses to decrease the risk of thrombosis and improve the clinical management in COVID-19 patients.

### 1.2. Hypercoagulability and Viral Infections

Hypercoagulability and vascular injury are stimulated by inflammation and hysterical viral replication [39]. In the Ebola viral infection, 30% of the patients showed hemorrhagic symptoms [40]. Wang et al., (2011) reported that patients with influenza A (H1N1) have low lymphocytes and high D-dimer levels. The significantly elevated D-dimer indicates the probability of the formation of a pulmonary microthrombus. Thus, they recommended that it may be necessary to consider anticoagulant therapy [41]. In addition, Borges et al., (2014) stated that D-dimer levels increased in HIV^+^ men and women and hepatitis B/C co-infected individuals [42]. Host inflammatory responses and/or antigen–antibody complexes of the virus can activate platelets, which are then cleared from the circulation by the reticuloendothelial system [43]. Platelets can be reduced by an interaction between megakaryocytes and viruses [44].

However, severe acute respiratory syndrome-COV-1 (SARS-CoV-1) has been associated with a low prevalence of bleeding. It caused pulmonary embolism, induced thrombocytosis, thrombocytopenia, long PT, and activated partial thromboplastin time (aPTT) [45,46]. It is noteworthy that COVID-19 may have a similar complication pattern to SARS-CoV-1. Interestingly, thrombocytopenia, which results from hypercoagulopathy in sepsis, has a relatively low prevalence in SARS-CoV-1 [47] and COVID-19 [1].

Thrombocytopenia in COVID-19 could be explained by continuous migration and consequent consumption of the platelets from the blood, owing to continuous inflammatory status [48]. The uncontrolled increase in the proinflammatory cytokines IL-6 and IL-1, which induce “cytokine storm” and can stimulate megakaryocyte proliferation, probably causing thrombocytosis.

The invasion of coronavirus through the airway causes tissue injury and elicits innate immune responses wherein the viral particles stimulate the activation of resident alveolar macrophages and complement cascade through the lectin pathway. Upon complement activation, the membrane attack complex can directly cause endothelial cell damage [49]. Furthermore, TF becomes exposed to external surfaces and comes into contact with coagulation factor VII, thereby activating the extrinsic coagulation pathway. Leukocyte infiltration and TF expression in inflammatory monocytes further worsen thrombotic complications. The generation of proinflammatory cytokines and vascular inflammation is mediated by several pattern recognition receptors, known as toll-like receptors (TLRs) and nod-like receptors (NLRs). In addition, activation of TLRs mediated by oxidized phospholipids and damaged cells’ hypoxic conditions caused by viral infection leads to monocyte infiltration and activates the production of TNF-α, IL6, IL-8, and interferon-c (INF-c), resulting in massive vascular endothelial and alveolar epithelial cell damage [50]. TLR3 can also cause hypercoagulation by activating TF expression [51]. Thus, coagulation system activation could be a risk factor for adverse outcomes of viral infection and, consequently, COVID-19 and should be taken into account in clinical practice.

### 1.3. Hypercoagulability Markers in COVID-19

There is a correlation between activation of the coagulation system and poor prognostic cases of severe infection of COVID-19 [10]. Currently, D-dimer, FDP, and PT are among the most important hypercoagulability markers in the COVID-19 pandemic. D-dimer is elevated in more than 45% of patients and is considered an independent risk factor for mortality in COVID-19 [38,52]. Patients with D-dimer level greater than 1000 ng/mL are at risk of death 20-fold more than patients with lower levels [38]. A retrospective study on 1099 patients, admitted to 552 hospitals all over China, showed that D-dimer is strongly elevated in patients with severe COVID-19 (59.6% of the patients) compared to less severe cases (43.2% of the patients) [1].

Platelets are another hypercoagulability parameter that was found to have decreased in the blood (thrombocytopenia) of severe COVID-19 cases (57.7% of the patients) in a retrospective study that used 1338 patients in many hospitals all over China [1]. However, a retrospective analysis in Tongji hospital showed that there was a high platelet count in patients with severe pneumonia owing to COVID-19 compared with non-COVID-19 ones [53].

A study was performed on a group of 22 patients suffering from acute respiratory failure and admitted to the ICU in the Hospital of Padova University after COVID-19 infection. The result showed that patients had significantly increased plasma levels of fibrinogen compared to healthy control patients [54]. A small thrombus containing fibrin/platelet was found in the pulmonary parenchyma and microcirculation after the development of ARDS in COVID-19 infected patients [55].

Furthermore, von Willebrand factor (VWF) and factor VIII activities are significantly increased in COVID-19 patients [56]. There was an increase in factor VIII and VWF in a study performed by Panigada et al., (2020) on 24 COVID-19 patients admitted to ICU [57]. In some studies, ARDS patients with associated disease severity, multiple organ failure, and mortality showed increased levels of plasminogen activator inhibitor-1 (PAI-1) and soluble thrombomodulin and decreased level of protein C [58,59,60,61,62]. In addition, COVID-19 patients developed ARDS with an increased level of TF in alveoli and plasma compared to pulmonary edema patients without COVID-19 [63]. High mortality rates owing to thrombotic complications are considered an important consequence of COVID-19 infection; therefore, approaches to inhibit thrombosis in COVID-19 patients are very important. Antithrombotic drugs such as heparin, fibrinolytics, dipyridamole, FXII inhibitors, and nafamostat have pleiotropic anti-inflammatory or antiviral effects [64].

Several capillary and vascular diseases, for example, hypercoagulability, microangiopathy, and venous/arterial thromboembolic crises are commonly associated with the natural COVID-19 infection, especially in moderate to severe cases [65]. Meanwhile, adenoviral vector-based vaccines can bind platelets and induce their destruction in the reticuloendothelial system. In addition, liposomal mRNA-based vaccines may activate the coagulation factors and provide a pro-thrombotic phenotype to endothelial cells and platelets [66]. Due to an imbalance in clinical trials, hypercoagulability is a risk management plan (RMP) and an important potential risk associated with the administration of the recently approved COVID-19 Vaccine Janssen, another adenovirus vaccine. The high level of systemic inflammatory response (SIR) within severe COVID-19 cases is a potential assumed mechanism that is related to the hypercoagulable statistics among patients [67]. Moreover, there is an association between adenoviral vector or mRNA-based COVID-19 vaccines and thrombotic events, resembling antiphospholipid syndrome (APS), which has appeared in few cases of thrombocytopenia after vaccination. This association appeared through the trigger of type I interferon response, which is associated with the generation of antiphospholipid antibodies (aPLs). aPLs may directly cause the activation of immune response and participation of innate immune cells, cytokines, and complement cascade. Therefore, aPLs that are linked with the risk of APS might represent a risk factor for thrombotic events following COVID-19 vaccination [68]. The major issue of SARS-CoV-2 infection without prior vaccines and immunity is the pathophysiologic responses associated with acute infection—hypercoagulability and thrombo-inflammation—which might drive the disease process and ongoing worldwide crises [69].

### 1.4. Fibrinolysis in COVID-19 Patients

Plasmin, a serine protease, is the main component in the fibrinolytic system. It results from the activation of plasminogen into plasmin [70]. Fibrinolysis is activated by tissue type plasminogen activator (tPA) and urokinase-type plasminogen activator (uPA). They bind to their specific receptors on cell surfaces. In order to prevent the over activation of fibrinolytic process, serine protease inhibitors (serpins) regulate fibrinolysis at various activation sites. Those inhibitors include activated protein C inhibitor (APC) inhibitor (PAI-3), plasminogen activator inhibitor 1 (PAI-1), plasminogen activator inhibitor 2 (PAI-2), defensin, and protease nexin 1, which inhibit the conversion of plasminogen to plasmin [71]. Suppression of fibrinolysis is exhibited frequently in cases of acute lung injury (ALI) [72,73]. Inhibition of fibrinolysis is considered an important provoking factor causing thrombotic complications that increase the severity of COVID-19. In sepsis, it is associated with disease progression, cell injury, and increased mortality rate [74,75]. Elevated thrombin activatable fibrinolysis inhibitor (TAFI) and PAI-1 were found to be a leading cause of hypofibrinolysis in COVID-19 patients [14]. In the acute phase of COVID-19 infection, influx of the coagulation factors and fibrinogen containing inflammatory fluids leads to formation of hyaline membrane and fibrin accumulation. Furthermore, increased inflammatory cytokines, such as IL-1, IL-6, and IL-17A, increase PAI-1 and inhibit uPA expressions [76,77]. A hypofibrinolysis then occurs, leading to not only increased fibrin accumulation and hyaline membrane formation but also a state of microvascular thrombosis. After prolonged COVID-19 infection, epithelial cells pass through an epithelial–mesenchymal transition [78] and consequently lead to fibrosis [79]. Therefore, it is worth noting that the degree of fibrinolytic state depends accordingly upon the severity of COVID-19.

### 1.5. Therapeutic Approaches for Hypercoagulability in COVID-19

Most of the complications of COVID-19 are DIC and hyperactivity of the host immune response [80]. Increased blood coagulation markers in patients with ARDS and sepsis are mostly associated with bad prognoses and poor outcomes. Therefore, anticoagulants such as heparin have been used clinically to target the coagulation system [81]. A recent retrospective study revealed the prophylactic or therapeutic benefits of heparin in patients with severe COVID-19 infection. Heparin may improve the situation through its anticoagulant and anti-inflammatory effects [37]. The International Society on Thrombosis and Haemostasis (ISTH) and the American College of Cardiology (ACC) have recently recommended a prophylactic calculated dose of heparin with low-molecular-weight (LMWH) or unfractionated heparin in all severe cases of COVID-19 patients [82,83]. Heparin is suggested to be used at a dose of 50 UI/Kg and in patients with bleeding or with thrombocytopenia [84].

Careful selection of appropriate anticoagulant drug(s) and dosage in addition to close observation of patients are very important, especially in patients with chronic diseases such as diabetes, hypertension, and liver disease. Therefore, special care must be taken with regards to patients with inherited related coagulation diseases such as von Willebrand hemostasis and hemophilia A or B.

### 1.6. Natural Products as Anticoagulants

All characteristics of the coagulation cascade, primary hemostasis, coagulation, and fibrinolysis can be affected during viral infections; therefore, thrombosis and DIC may occur. During severe viral infections, nothing can be used except supportive treatment [85]. After vascular damage, the integrity of the high pressured closed circulatory system is preserved by the hemostatic system. Normally, thrombi formation is controlled by a regulatory system; however, under pathological conditions or a shift in the hemostatic balance toward the procoagulant side, thrombi formation is initiated [86]. Hemostasis disorders are a serious manifestation in COVID-19 patients. Thrombotic complications are among the most important causes of death in patients with severe COVID-19 infection. Many research programs are directed toward finding new effective and safe antithrombotic agents. In this regard, natural products present a generous source for promising candidates because they are cheaper and less toxic than synthetic drugs. Accordingly, they can offer important complementary drugs for recovering from hemostasis disorders in COVID-19 patients. Table 1 shows different natural sources, either in the form of extracts or isolated pure compounds, that have been reported to possess antithrombosis activity. Chen et al., (2015) reviewed natural products on the basis of their antithrombosis activity. They classified the drugs into antiplatelet aggregation, anticoagulant, and fibrinolytic acting drugs [87]. Anticoagulant drugs were further classified into TF inhibitors and inhibitors of the coagulation pathway. Likewise, antiplatelet drugs were subdivided into four main subclasses: drugs that act as inhibitors of platelet membrane receptors, drugs that impact on the nucleotide system, inhibitors of platelet granules secretion, and drugs that impact on the arachidonic acid system (Table 1). A wide variety of natural products, primarily from plants, were proved to exhibit a significant fibrinolytic activity, e.g., gingko, ginger, raspberries, garlic, onion, fermented soybeans, *Ananas comosus*, *Fagonia arabica*, *Spirodela polyrhiza*, *Flammulina velutipes*, *Lagenaria siceraria*, *Bacopa monnieri*, *Clausena suffruticosa*, *Leea indica*, *Leucas aspera*, *Pinus densiflora*, *Lonicera japonica*, *Sargassum fulvellum*, *Pueraria lobata*, *Trichosanthes kirilowii*, *Lonicera japonica*, and *Desmodium styracifolium* [88].

Nowadays, marine organisms receive considerable interest as a renewable source for drugs. During the last few years, several antithrombotic drugs have been reported from marine sources. Marine polysaccharides represent an important class of marine-derived antithrombotic agents. Kuznetsova et al., (2021) reviewed the potency of seaweed sulfated polysaccharides for correcting hemostasis disorders in COVID-19 [89]. Moreover, other molecules from different chemical classes, such as peptides, polyketides, steroids, terpenes, alkaloids, and polyphenols, have proven antithrombosis efficacy in in vitro and in vivo assays. Some secondary metabolites such as terpenes (dichotomanol, dolastane diterpene, and pachydictyol), protein (YAP), and polyphenol (phloroglucinol) exhibited anticoagulant and antiplatelet activities [90].

Therefore, it is clear that natural products, whether from terrestrial or marine sources, are a diverse source of anti-thrombotic agents. In this context and on the basis of the diversified mechanisms of action of natural products, these agents could be considered as a complement to the currently used anti-thrombotic drugs. Hence, it can be one of the recommended medications to reduce deaths from COVID-19 infection. Nevertheless, to date, no research findings regarding the investigation of the use of natural anticoagulants as supportive therapy for COVID-19 infection have been published.

**Table 1 viruses-14-00228-t001:** Natural sources (for extracts or pure compounds) used as antithrombosis and their mechanisms of action.

Mechanism of Action		Natural Source	Active Constituents	References
Anticoagulant drugs	1-TF inhibitors	*Chaenomeles sinensis*	Hovertrichoside, luteolin-7-O-β-D-glucuronide, hyperin, avicularin and quercetin	[91]
		*Ligustici chuanxiong*	Ligustrazine	[87]
		*Eriobotrya japonica* Lindley	Sesquiterpene glycoside	[92]
		Beans and grain	α-Zearalanol	[87]
	2-Inhibitors of the intrinsic and extrinsic coagulation pathways	The green algae *Monostroma arcticum*	Polysaccharide	[87]
		*Polygala fallax* Hesml.	Saponins	[87]
		*Rhododendron brachycarpum*	Hyperoside,	[93]
		*Umbilicaria esculenta*	Polysaccharide	[94]
		*Withania somnifera*	Withaferin A	[95]
		*Scutellaria baicalensis* Georgi	Wogonin and wogonoside	[96]
		*Erigeron canadensis* L.	Polyphenolic-polysaccharide preparation	[97]
		*Codium vermilara*	Polysaccharide	[98]
		*Crassocephalum crepidioides*	Crude extract	[99]
Anti-platelet aggregation drugs	1-Acting by variable mechanisms	*Andrographis paniculata*	Andrographolide	[100]
		*Bupleurumfalcatum*	Bupleurumin	[101]
		*Salvia milthorriza* Bunge	Tanshinone IIA	[102]
		*Abies webbiana*, parsley, *Nigella sativa*	Crude extract	[103,104,105]
	2-Inhibitors of platelet membrane receptors	*Spatholobus suberectus,* garlic	Crude extract	[106,107]
		*Rabdosia japonica* var. *glaucocalyx*	Glaucocalyxin A	[108]
		*Salvia miltiorrhiza*	Salvianolic acid B	[109]
		*Garcinia nervosa* var. *pubescens* King	Flavonoids	[110]
		*Erylus formosus*	Eryloside F	[111]
		*Piper longum*	Piperlongumine	[112]
		*Licania pittieri*	Pomolic acid	[113]
		*Polygonum multiflorum*	Tetrahydroxystilbene glucoside	[114]
		*Agkistrodon acutus* Venom	Tripeptide	[115]
		Cruciferous vegetables	Indole-3-carbinol	[116]
		*Goniothalamus species*	Essential oils	[117]
		*Ligusticam wallichii Franch*	Tetramethyl pyrazine	[118]
		*Agrimonia pilosa, Toona sinensis*	Crude extract	[87]
		*Rhus verniciflua Stokes*	Isomaltol and pentagalloyl glucose	[119]
	3-Impacting on nucleotide system.	*Cordyceps militaris*	Cordycepin	[120]
		*Ginkgo biloba*	Ginkgolide C, quercetin	[121]
		*Oligoporus tephroleucus*	Oligoporin A	[122]
	4-Inhibitors of platelet granules secretion.	Saffron	Crocetin	[123]
		Black soybean	Crude extract	[124]
		Magnolia bark	Magnolol	[125]
		*Solanum lycopersicum*	Guanosine	[126]
		*Ligustici Chuanxiong*	Ligustrazine ferulate,	[87]
		*Rhizoma Curcumae*	Curdione	[127]
	5-Impacting on arachidonic acid system	Green tea leaves	Epigallocatechin-3-gallate	[128]
		*Zizyphus jujube*	Jujuboside B	[129]
		Sorghum vinegar	Alditol and monosaccharide	[130]
		*Magnolia obovate*	Diacetylated obovatol	[131]
		*Artemisia princeps* Pampanini	Crude extract, eupatilin, and jaceosidin	[132]
		Grape fruits and oranges	Hesperetin	[133]
		Betel leaf	Hydroxychavicol	[134]
		*Stephaniae tetrandrae*	Tetrandrine and fangchinoline	[135]
		*Uncaria sinensis* (Oliv.) Havil.	Isorhynchophylline	[87]
		*Caesalpinia sappan* L.	Ethyl acetate extract	[87]
		*Cornus officinalis* Sieb. et Zucc	Morroniside	[87]
		*Pleurothyrium cinereum*, *Ocotea macrophylla* and *Nectandra amazonum*	Neolignans	[136]
		*Zingiber mioga* Roscoe	Crude extracts	[137]
		White ginseng	Ginsenoside Rk1	[138]

### 1.7. Nanotechnology for Future Treatment of COVID-19

Nanomaterials are effective tools in the field of medicine because they can target highly specific cells and avoid the side effects of medical drug use. In particular, nanoparticles (NPs) can bind with ligands to target cell surfaces. NPs can be conjugated with ligands for active targeting; these ligands could be peptides, antibodies, or hormones [139,140]. This kind of conjugation could be beneficial for destroying viruses such as COVID-19. Through the management of size; surface features; and the material used as smart systems by encasing, enclosing, and coating the drugs and imaging agents, nanoparticles can be used in drug delivery. The potential uses of anticoagulation in COVID-19 could be of interest when conjugated with NPs. For example, heparin as an anticoagulant can be conjugated to NPs as an antiviral agent against COVID-19 (Figure 2) [141].

To date, the most promising approach to treating thrombosis using nanomedicine has been to deliver antithrombotic drugs to thrombus sites by targeting one or more proteins involved in coagulation (for example, fibrin, thrombin, or hydrogen peroxide (H_2_O_2_)) to the thrombus sites. A different approach, which has been documented previously, involves the targeting of cells such as active platelets involved in the coagulation process, with cell-binding ligands. Researchers were able to reduce the formation of H_2_O_2_, using an H_2_O_2_-responsive boronate antioxidant polymer (BAP) bonded to the fibrin-targeting lipopeptides. In rat models, Kang et al., 2017 demonstrated that nanoparticles could carry tirofiban to the thrombus site, reducing H_2_O_2_ production and hence TNF-α and soluble CD40 [142]. Both functionalized iron oxide nanoparticle micelles and bare nanoparticles were used to detect thrombus locations using magnetic particle imaging [143]. Superparamagnetic iron oxide nanoparticles coated with fucoidan, a polysaccharide with a high affinity for activated platelets, have been demonstrated to connect to P-selectin and aid the detection of thrombus formation in vivo, according to researchers [144]. Additionally, liposome nanoparticles containing cyclic Arg-Gly-Asp (RGD) on their surfaces are effective at targeting the activated platelet receptor integrin GPIIb-IIIa [145]. Park et al., (2006) produced an amphiphilic combination of heparin and deoxycholic acid that encapsulated doxorubicin in a two-step method for SCC (squamous cell carcinoma). Therefore, nanoparticles were examined for cytotoxicity, antitumor activity, and toxicity. The conjugate has high loading and release efficiency, promoting antitumor activity [146]. The heparin/DOX/DEVD-S-DOX complex was formed first by Khaliq et al., (2006). The composite was then stabilized using Pluronic F-68. When delivered to the tumor site, DOX (doxorubicin) was exposed to the tumor cells, prompting apoptosis and recurrent activation of caspase-3. Caspases are proteases that control cell death and inflammation—they cause apoptosis, which is triggered in nanoformulations. This study used SCC-7 murine squamous cell carcinoma cells [147].

Additionally, these systems can deliver anticoagulant drugs to specific tissues and ensure controlled-release therapy [148]. The anticoagulant drugs can be taken via inhalation, absorption throughout the skin, or injection. Smart nanocarriers comprised organics (polymeric micelles, liposomes, and hydrogels), which can enclose anticoagulant drugs inside, and inorganics (quantum dots, gold, and silica nanoparticles), which can carry the anticoagulant on their surfaces [149]. Lembo et al. [150] developed heparin nanoassemblies based on the self-association of O-palmitoyl-heparin and a-cyclodextrin in water for antiviral activity against herpes simplex viruses, human papillomavirus, and respiratory syncytial virus. Joshi et al. [151] conjugated chloroquine as an antiviral drug with thiol-functionalized gold nanoparticles, and the chloroquine-conjugated gold nanoparticles interacted well with bovine serum albumen and showed antiviral activity. Hsu et al. [152] used berberine, a natural isoquinoline alkaloid with antiviral activity, and then formulated it as novel berberine nanoparticles comprising heparin and shelled with linear polyethyleneimine.

The SARS-CoV-2 virus that produces COVID-19 disease enters the cells by targeting receptors of angiotensin-converting enzyme 2 (ACE2). Because the virus has ACE2 receptors, the NPs can also carry anticoagulants and conjugate with ACE2 ligands to bind to receptors of the virus cell. Figueroa et al. [153] developed NP-coated angiotensin-converting enzyme on their surface using angiotensin for targeting viruses. The presence of the receptor on the surface of the viruses confirmed NP uptake via endocytosis. The results proved that the design of virus-mimetic cell interactive NPs is a precious strategy for improving NP specificity for therapeutic and diagnostic applications. Itani et al., (2020) suggested the development of theranostic NPs for the professional and selective delivery of beneficial moieties (that is, drugs such as anticoagulants, vaccines, siRNA, peptides) to actively target sites of infection and help in the fight against COVID-19 [154]. They suggested this kind of NPs because intranasal delivery was the favorite administration route against viral pulmonary diseases. The theranostic NPs can be divided into three broad categories: (1) the organic category such as liposomes, polymeric nanoparticles, and dendrimer; (2) inorganic category, such as gold, silver, quantum dots, and iron nanoparticles; and (3) virus-like or self-assembling protein nanoparticles [148]. Therefore, the formulation of the currently used anti COVID-19 drugs that have anti-hypercoagulability in a nanoparticle form appears to be a promising strategy in the future.

Unusual thrombosis and thrombocytopenia have been identified as uncommon side effects linked with the ChAdOx1 nCov-19 (AstraZeneca) COVID-19 vaccination. The underlying causes of this uncommon disease are unknown; however, a clinical similarity to heparin-induced thrombocytopenia (HIT) has been noticed. Eichinger and colleagues examined individuals who acquired a coagulation condition resembling HIT following immunization with ChAdOx1 nCov-19, even though they did not receive heparin prior to symptom onset. Plasma samples from patients indicated the presence of an antibody consistent with an autoimmune form of HIT, in which platelets become abnormally activated in the absence of heparin. The authors proposed a diagnostic and therapeutic strategy for treatment on the basis of test and clinical data [155,156]. Furthermore, current research shows that 20–30% of patients at high risk of COVID-19 mortality develop blood clotting, resulting in stroke and sudden death. Identifying the degree of blood clotting can help clinicians choose the best blood thinners to avoid life-threatening complications. Rapid detection of clotting-related proteins in COVID-19 plasma could save countless lives. Patients at high risk of death from COVID-19 infections, including blood clots, are being identified using nanotechnology. With improved mass spectrometry-based proteomics methods, nanomedicine can discover critical protein patterns that are involved in the occurrence and course of this disease. The combination of these sophisticated techniques may help us better understand clotting and develop new diagnostics and therapies for COVID-19 [157].

### 1.8. Nanoparticle-Loaded Anticoagulants

Several anticoagulant-based nanoformulations for delivery systems have been investigated to increase their therapeutic efficacy while also decreasing their toxicity when administered through diverse routes. Whatever the drug delivery system (DDS) is used, the evaluation of anticoagulant performance is essential for the creation of a DDS that is appropriate for patients. In both scientific research and clinical practice, it is critical to have a thorough grasp of the many anticoagulant tests available, as well as their main uses and limits. This is partly owing to the use of nanoscale formulations in anticoagulant treatment. In recent years, the limits to the use of anticoagulants have been lessened [158].

Table 2 has a large number of formulations using various types of nanoparticles that were found to be of appropriate size for dispersion in the body. LMWH-based nanocarriers with an emphasis on anticoagulation performance are created with various and acceptable nanosizes, with the anticoagulation performance of the nanocarriers being the most important consideration. The use of conventional anticoagulants has been around for decades in the treatment of a wide variety of illnesses [159]. Compared to heparin and vitamin K antagonists, direct oral anticoagulants have fewer side effects. A key difficulty connected with these anticoagulants continues to be the lack of a proper laboratory assessment and the absence of a factor “xaban” (Xa) inhibitor reversal medication [160]. LMWHs are anticoagulant agents with excellent efficacy and safety profiles, and they have been investigated for improving outcomes by incorporating various nanocarriers into their formulations and administering them through a variety of administration routes, including intravenous administration [161]. With the use of nanoformulations, it is possible to increase the efficiency of LMWHs by overcoming the major issues that have arisen in the past.

### 1.9. COVID-19 Vaccine Loaded Nanoparticles

The FDA gave the first emergency use permit ever given to a coronavirus vaccine by the United States [171,172]. BioNTech researchers initiated work on the vaccine in January 2020, on the basis of a genetic molecule called messenger RNA (mRNA). The vaccine includes genetic instructions, known as a spike, for developing a coronavirus protein. The vaccine allows them to produce spike proteins when inserted into cells and are then released into the body and thus provoke an immune system response. Pfizer-BioNTech’s vaccines are formed in liposomal nanoparticles (LNPs) or PEGylated liposomes (PEGLip). PEGlip are artificial phospholipid vesicles that are effective in stabilizing pharmaceutical products and enhancing their pharmacological properties, and in the case of these COVID-19 vaccines, this liposome formulation enables mRNA to be stabilized, owing to its lability. Moreover, Moderna produces the vaccine from mRNA, but they are yet to market one. Moderna are formulated in PEGylated liposomes (PEGLip). PEGlip are artificial phospholipid vesicles that have proven to be useful in stabilizing drugs and improving their pharmacological properties. Furthermore, the Oxford–AstraZeneca vaccine is based on the genetic instructions of the virus for the construction of spike protein. However, the Oxford vaccine uses double-stranded DNA, unlike the Pfizer–BioNTech and Moderna vaccines, which store the instructions in single-stranded RNA. The researchers attached another virus called adenovirus to the gene for the coronavirus spike protein. Novavax COVID-19 vaccine, also known as Nuvaxovid and Covovax [173] is a subunit COVID-19 vaccination developed by Novavax and the Coalition for Epidemic Preparedness Innovations (CEPI). Nuvaxovid’s crucial phase III study data were published in its entirety in December 2021 [174]. NVX-CoV2373 has been characterized as a vaccine against both protein subunits and virus-like particles. [175,176], although the producers call it a “recombinant nanoparticle vaccine”. The vaccine is made by engineering a baculovirus with a gene encoding a modified SARS-CoV-2 spike protein. Two proline amino acids were added to the spike protein to stabilize the pre-fusion version of the protein; this same 2P modification is employed in various other COVID-19 vaccines [177]. A saponin-based adjuvant is included in the formulation [178,179] (Table 3).

### 1.10. Coagulation System Activation and COVID-19 Vaccines

Many vaccines were manufactured after extensive research on COVID-19 infection, and these vaccines have spread all over the world [180,181,182]. Moreover, regarding the uncommon side effects following the use of COVID-19 vaccines such as anaphylaxis [183], new reports have found thrombosis with thrombocytopenia syndrome (TTS)-associated venous and arterial thromboembolism [184]. Two common vaccines associated with TTS are Oxford–AstraZeneca or Vaxzevria (ChAdOx1nCoV-19) and Johnson and Johnson (AD26.COV2•S) [185]. Both vaccines contain chimpanzee (ChAdOx1nCoV-19) or human (AD26.COV2•S) recombinant adenovirus vectors that encode the skeletal protein of SARS-CoV-2 [186]. In March 2021, several European countries stopped providing the Oxford–AstraZeneca vaccine due to TTS concerns [187]. Reports have shown that among the 34 million people vaccinated by Oxford–AstraZeneca in Europe, 169 cases of cerebral venous thrombosis and 53 cases of visceral venous thrombosis with TTS were recorded [188]. In addition, in the USA, the FDA suspended Johnson and Johnson vaccines in April 2021 due to TTS concerns [189]. Out of 6.8 million people vaccinated by Johnson and Johnson in the USA, 15 cases of thrombosis have been reported. However, no cases of TTS have been described using mRNA vaccines such as Pfizer–BioNTech and Moderna [190]. Considerably, the European Medicines Agency and the Centers for Disease Control and Prevention (CDC) have reported that the benefits of the Oxford–AstraZeneca and Johnson and Johnson vaccines outweigh their risks, including TTS [190]. Moreover, the efficacy of AstraZeneca vaccine in terms of the strong reduction in hospitalizations and death caused by COVID-19 infection, according to the European Medicines Committee (EMA), far outweighs the potential for TTS [191]. Although the mechanism of TTS is not yet known, it is likely to be through the formation of antibodies that act against platelet antigens, resulting in the activation and a large aggregation of platelets, which reduces the number of platelets and leads to thrombosis [187]. Another hypothesis is this: when the adenovirus reaches the blood, it stimulates a conflicting immune response, which leads to platelet activation and decreases the level of ACE2 enzyme on the surface of endothelial cells. Platelet activation could also lead to NET, which results in augmented thrombotic threat [192].

Fever, muscle pain, fatigue, and headache are the early symptoms of COVID-19 vaccines [192]. TTS should be considered in each of the following: headache persisting for more than 3 days, leg swelling, abdominal pain, shortness of breath, chest pain, leg pain, or vomiting after receiving either Oxford–AstraZeneca or Johnson and Johnson vaccines [2]. Complete blood count (CBC), D-dimer, fibrinogen, coagulation panel, and PF4-heparin ELISA are the most important diagnostic tools for TTS cases [190]. To date, important indications in reported cases are increased levels of PF4-heparin ELISA and D-dimer with decreased platelet count and fibrinogen levels [192]. It is advisable to consult with hematologists when there is a suspicion of the presence of TTS in the persons receiving the vaccine.

## 2. Conclusions

In conclusion, hypercoagulability is one of the risk factors that lead to poor outcomes of viral infections in general, and in COVID-19 in particular. Furthermore, hypofibrinolysis is a common feature in patients with COVID-19 infection, and the degree of fibrinolysis inhibition depends on the severity of the infection. It is worth noting that the use carefully controlled doses of anticoagulant therapy, to reduce the risk hypercoagulability in COVID-19 patients, is considered one of the important positive practices. However, caution should be exercised when using anticoagulants in COVID-19 patients, especially those with hereditary coagulation disorders. Other important points to be considered are the promising anticoagulant potential of natural sources of drugs, which is attributable to their safety and diversified mechanisms of action, as well as the promising applications of nanoparticles technology in the design of COVID-19 vaccines and drugs.

### Compliance with Ethical Standards

The review proposal was approved by Al-Azhar University ethical committee and performed in accordance with the regulations of the international guide for the care and use of laboratory animals.

## Figures and Tables

**Figure 1 viruses-14-00228-f001:**
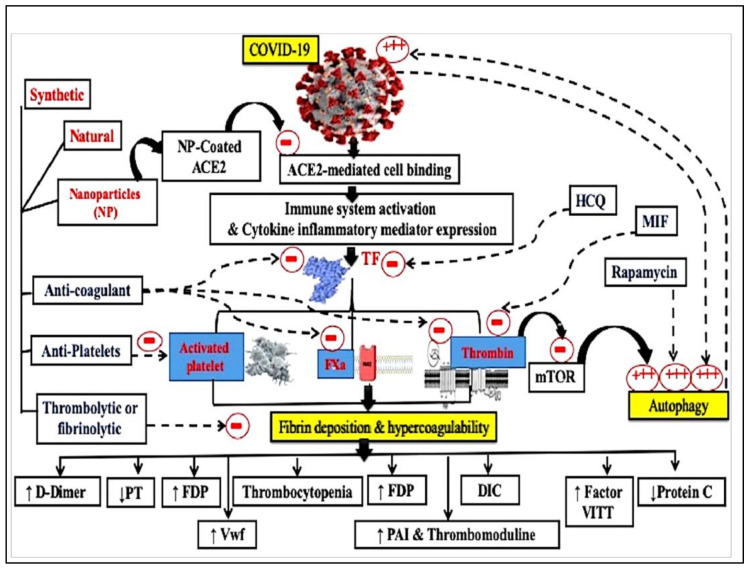
Schematic diagram representing the different coagulation system mechanisms and possible types of coating nanoparticles for targeting ang I and II receptors. DIC, disseminated intravascular co-agulation; Fxa, activated factor x; PT, prothrombin time; TF, tissue factor; Hydroxychloroquine—HCQ; Macrophage migration inhibitory factor—MIF; Fibrin degradation products—FDP; Von Willebrand factor—Vwf; Mammalian target of rapamycin—mTOR; plasminogen activator inhibitor—PAI.

**Figure 2 viruses-14-00228-f002:**
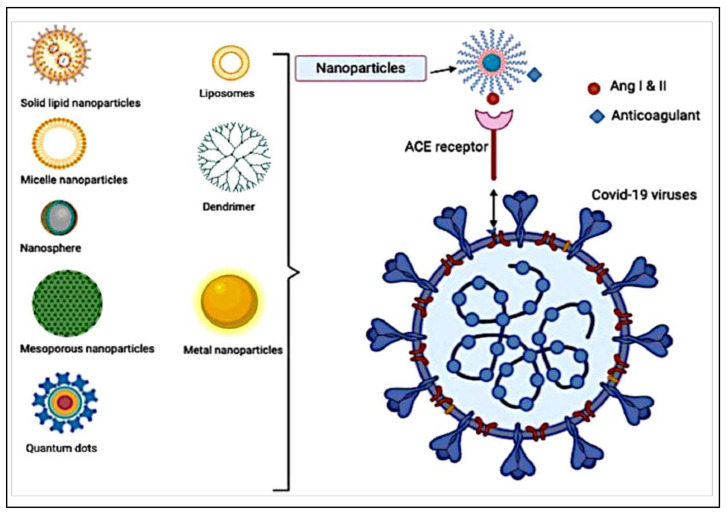
Schematic diagram representing the different formulated nanoparticles coated with anticoagulant for targeting ang I and II receptors.

**Table 2 viruses-14-00228-t002:** Nanoparticles loaded with anticoagulant therapies using different drug delivery systems.

Drugs	Types of Nanoparticles	Size Range (nm)	References
Low-molecular-weight heparin (LMWH)	Liposomes	80–90	[159]
Ardeparin (LMWH)		100–150	[162]
Enoxaparin (LMWH)		40–65	[163]
Unfractionated (UFH) heparin	Nanogel	130	[164]
Bemiparin (LMWH) Nadroparin (LMWH) Tinzaparin (LMWH)		150–400	[161]
Enoxaparin (LMWH)		100–1000	[165]
Enoxaparin	Polymeric nanoparticles	280–320	[166,167]
Fondaparinux		40–65	[168]
Enoxaparin		180–195	[169]
(LMWH)	Solid lipid nanoparticles	280–380	[170]
Enoxaparin (LMWH)	Self-nanoemulsifying drug delivery system	30–245	[167]
Rivaroxaban (Factor Xa inhibitor)		50–150	[160]

**Table 3 viruses-14-00228-t003:** Characteristics of the Pfizer/BioNTech, Oxford University/AstraZeneca, and Moderna vaccines [19,179].

Characteristics	Pfizer/BioNTech	Oxford University/AstraZeneca	Moderna	Nuvaxovid and Covovax
Therapeutic indication	For effective immunization to suppress SARS-CoV-2 virus-induced COVID-19 in persons 16 years of age and over.	For effective immunization for the prevention of COVID-19 in persons 18 years of age and over.	For effective immunization to prevent SARS-CoV-2 virus-induced COVID-19 in persons 18 years of age and over.	The vaccine is administered in two doses and is stable at refrigerated temperatures of 2 to 8 °C (36 to 46 °F).
Type of vaccine	Messenger RNA (mRNA)	Adenovirus vector	Messenger RNA (mRNA)	Recombinant nanoparticle vaccine
Number of doses	A multidose vial	One dose	Multidose	Multidose
Pharmaceutical form	Concentrate for solution for injection.	Solution for injection.	Dispersion for injection.	Dispersion for injection.
Dosage schedule	Two doses (0.3 mL each) with an interval of between 3 to 12 weeks.	Two doses (0.5 mL each) with an interval of between 4 and 12 weeks.	Two doses (0.5 mL each). It is recommended that the second dose be administered 28 days after the first dose.	The vaccine requires two doses and is stable at 2 to 8 °C (36 to 46 °F) refrigerated temperatures.

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
