# Peer review of "Coagulation System Activation for Targeting of COVID-19: Insights into Anticoagulants, Vaccine-Loaded Nanoparticles, and Hypercoagulability in COVID-19 Vaccines"

_viruses, 2022, doi:10.3390/v14020228_

Round 1

Reviewer 1 Report

Introduction is not very well organised 

Paragraph 1.1 (except for figure) to paragraph 1.6 do not give sufficient new contribution to the field and are repetitive

Paragraph 1.7 (natural products...) may be better explained and corroborated by scientific evidence.

Paragraph 1.10 and 1.11 do not give new contribution to the understanding of the subject.

Conclusion is poor.

Author Response

30 Dec 2021

J Viruses

Dear editor,

The authors would like to thank and appreciate the efforts of the editor and reviewers in the processing of our manuscript. We thank you and the reviewers for your thoughtful comments on the manuscript and have edited it to address these concerns. We declare that all reviewers’ comments have been taken in consideration and implemented in the manuscript (*All changes inside the manuscript have been done in red font). Below is a point-by-point response to the reviewers` comments. Moreover, the article was edited by a native language speaker, please see the attached certificate.

Reviewers Comments

Authors response

Reviewer 1

Introduction is not very well organized 

The introduction has been revised and organized 

Paragraph 1.1 (except for figure) to paragraph 1.6 do not give sufficient new contribution to the field and are repetitive

Paragraphs from 1.1-1.6 have been updated and modified according to the reviewer comment (see page 4-12)

Paragraph 1.7 (natural products...) may be better explained and corroborated by scientific evidence.

The section has been improved (see page 12)

Paragraph 1.10 and 1.11 do not give new contribution to the understanding of the subject.

Paragraph 1.10 and 1.11 have been updated and modified according to the reviewer comment (see page 18-20)

Conclusion is poor.

Conclusion has been improved

We look forward to working with you and the reviewers to move this manuscript closer to publication in J of Viruses.

Sincerely,

Ahmed A. H. Abdellatif

Associate Professor of Pharmaceutics, College of Pharmacy,

Qassim University, Saudi Arabia

E-mail address: a.abdellatif@qu.edu.sa

Reviewer 2 Report

The paper deals with an exciting topic. I believe that it should be significantly shortened.

Furthermore,  I suggest authors should make the clinical message (s) clear and highlight to readers 

The paper should undergo English editing to improve fluency and readability.

Author Response

30 Dec 2021

J Viruses

Dear editor,

The authors would like to thank and appreciate the efforts of the editor and reviewers in the processing of our manuscript. We thank you and the reviewers for your thoughtful comments on the manuscript and have edited it to address these concerns. We declare that all reviewers’ comments have been taken into consideration and implemented in the manuscript (*All changes inside the manuscript have been done in red font). Below is a point-by-point response to the reviewers` comments. Moreover, the article was edited by a native language speaker, please see the attached certificate.

Reviewers Comments

Authors response

Reviewer 2

The paper deals with an exciting topic. I believe that it should be significantly shortened.

Manuscript has been shortened

Furthermore, I suggest authors should make the clinical message (s) clear and highlight to readers 

Clinical messages have been introduced to all manuscript sections

The paper should undergo English editing to improve fluency and readability.

The manuscript has been improved in term of  fluency and readability

We look forward to working with you and the reviewers to move this manuscript closer to publication in J of Viruses.

Sincerely,

Ahmed A. H. Abdellatif

Associate Professor of Pharmaceutics, College of Pharmacy,

Qassim University, Saudi Arabia

E-mail address: a.abdellatif@qu.edu.sa

Reviewer 3 Report

Generally, the work is written carefully. Nevertheless, its elements require significant corrections.

  1. What is the purpose of the description in the introduction? The authors give various concepts, but they can be confusing without developing them.
  2. Are the authors sure that 15% of COVID-19 patients require hospitalization at the ICU?
  3. Classic DIC is not found in COVID-19.
  4. The authors mention hypercoagulability in COVID-19 without writing about fibrinolysis. This is a massive oversight (DOI: 10.1055/a-1346-3178).
  5. What is the purpose of describing the changes observed in other viral diseases? It introduces complete chaos.
  6. What is also the purpose of describing hypercoagulability as a separate subsection? The description is relatively trivial, incomplete, and not based on the latest reports.
  7. The figures are of poor quality and should be described in more detail.
  8. COVID-19 viral load and autophagy - additional elements of the manuscript that seem very random, with no significant relationship to the paper's main topic.
  9. Natural anticoagulants - a chapter that is not justified in any study. The authors' conclusions are too far-reaching and careless.
  10. The following descriptions are authors' conclusions with no literature support. These are hypotheses; hence it is challenging to consider the entire manuscript as a narrative review on the subject of the title and abstract.

Author Response

30 Dec 2021

J Viruses

Dear editor,

The authors would like to thank and appreciate the efforts of the editor and reviewers in the processing of our manuscript. We thank you and the reviewers for your thoughtful comments on the manuscript and have edited it to address these concerns. We declare that all reviewers’ comments have been taken into consideration and implemented in the manuscript (*All changes inside the manuscript have been done in red font). Below is a point-by-point response to the reviewers` comments. Moreover, the article was edited by a native language speaker, please see the attached certificate.

Reviewers Comments

Authors response

Reviewer 3

Generally, the work is written carefully. Nevertheless, its elements require significant corrections.

Required corrections are done

What is the purpose of the description in the introduction? The authors give various concepts, but they can be confusing without developing them.

The introduction includes an overview of COVID-19, its symptoms, and the mechanism of infection. It also describes the relationship between the coagulation system activation and the infection with COVID-19 in order to briefly introduce the subsequent parts of the review.

Based on the reviewer`s important comment, we updated and simplified the introduction section.

Are the authors sure that 15% of COVID-19 patients require hospitalization at the ICU?

This sentence has been updated (see introduction section)

Classic DIC is not found in COVID-19

DIC was connected to the increased mortality in COVID-19 pneumonia (Zhou X et al., 2021) however,  DIC is not common among COVID-19 patients (Wool GD, Miller JL., 2021). Please see an updated sentence in the  introduction section(please see page 3)

The authors mention hypercoagulability in COVID-19 without writing about fibrinolysis. This is a massive oversight (DOI: 10.1055/a-1346-3178).

A new section about fibrinolysis in COVID-19 has been involved (please see;

“Fibrinolysis in COVID-19 patients”

What is the purpose of describing the changes observed in other viral diseases? It introduces complete chaos.

This part describes the activation of the coagulation system in case of infection by many viruses, which has been previously confirmed in several studies. Similarly, this activation occurs also resulted in COVID-19. Therefore, we believe that it is worth starting with this part, which provides better flow and inclusion to different review parts.

What is also the purpose of describing hypercoagulability as a separate subsection? The description is relatively trivial, incomplete, and not based on the latest reports.

Because we are reviewing the relationship between coagulation system activation during COVID-19 infections, it is reasoning to give brief information about hypercoagulability, in order to give the reader an introduction that facilitates understanding of the terms and concepts related to it.

The figures are of poor quality and should be described in more detail.

Figures have been improved (see figures)

COVID-19 viral load and autophagy - additional elements of the manuscript that seem very random, with no significant relationship to the paper's main topic.

The sections of “COVID-19 viral load” and autophagy have been removed

Natural anticoagulants - a chapter that is not justified in any study. The authors' conclusions are too far-reaching and careless.

Natural products represent a cornerstone in the drug discovery process. Since ancient times, disease treatment begins with the use of plant extracts. Till now exploration of the biological potential of natural products represents an important strategy for disease treatment. Much more, the safety issues add more advantages to the use of natural products.

 During the COVID-19 pandemic, plenty of research was directed toward the exploration of natural sources and their metabolites content as hope for finding inspiring chemical moieties that can help with this unexpected catastrophe.

After the emergence of the role of coagulation pathways in the death cases observed in severe complications of COVID-19, again natural products were considered as a promising candidate for finding anticoagulant drugs that can even work through different pathways.

Accordingly, we find that the review of previous research on Natural antithrombotic drugs could provide a helpful source for collective data about different chemical classes, their natural sources, and their mechanisms of action.

The following descriptions are authors' conclusions with no literature support. These are hypotheses; hence it is challenging to consider the entire manuscript as a narrative review on the subject of the title and abstract.

conclusions assumed by the authors and resulted in a misunderstanding have been removed

We look forward to working with you and the reviewers to move this manuscript closer to publication in J of Viruses.

Sincerely,

Ahmed A. H. Abdellatif

Associate Professor of Pharmaceutics, College of Pharmacy,

Qassim University, Saudi Arabia

E-mail address: a.abdellatif@qu.edu.sa

Round 2

Reviewer 1 Report

Page 8 and 20: I don't aeree with this statement, that should  be demonstrated. (However short-term clinical trials.....also within the normal).

Page 11: I don't aeree with this statement, that should  be demonstrated (or even as a substitute)

Page 18: add Novavax

Page 20: English may be improved (it is possibile to design nanoparticles....to treat illnesses caused by coagulation  system  imbalance)

Author Response

Dear Viruses journal editor,

The Authors would like to thank and appreciate the efforts of editor and reviewers in the processing of our manuscript. Below is point by point response to the reviewer’s comments.

Reviewe-1 Comments

  1. Page 8 and 20: I don't agree with this statement that should be demonstrated. (However short-term clinical trials.....also within the normal).

Response

We are agree with the reviewer comment therefore the sentences have been removed (see page 8 and 20)

  1. Page 11: I don't agree with this statement, that should be demonstrated (or even as a substitute)

Response

Statement has been modified (see page 11)

  1. Page 18: add Novavax

Response

Novavax has been added (see page 18)

  1. Page 20: English may be improved (it is possible to design nanoparticles....to treat illnesses caused by coagulation system  imbalance)

Response

English has been improved (see page 20)

Kind regards

Dr. Ahmed A. H. Abdellatif

Department of Pharmaceutics, College of Pharmacy, Qassim University, Qassim, 52471, Kingdom of Saudi Arabia

Mail: a.abdellatif@qu.edu.sa

Reviewer 3 Report

The authors made a few minor corrections to the manuscript. Nevertheless, the whole article is a collection of random topics unrelated to each other. The text is relatively trivial and gives the impression of being written by researchers who do not have enough experience in medical science. The job adds absolutely nothing to the state of the knowledge of COVID-19.

Author Response

Dear Viruses journal editor,

The authors would like to thank and appreciate the efforts of the editor and reviewers in the processing of our manuscript. Below is a point by point response to the reviewer’s comments.

Reviewer-3 Comments

  1. The authors made a few minor corrections to the manuscript. Nevertheless, the whole article is a collection of random topics unrelated to each other.

Response

In reply to the reviewer-3 comments that the review is not interconnection, we believe that our review covers a very important as well as a hot topic for the researchers in the field. We provide comprehensive information about the coagulation system activation, COVID-19 infection, current and proposed treatments of the hypercoagulability, natural products as a renewable source for new chemical moieties with diversified mechanisms of action and limited side effects,  and the modern nano-formulations of the anticoagulants and vaccines. We also, provide an important section for the relation between the current COVID-19 vaccination and the bad consequences that could occur concerning the coagulation system. Here are the detailed connection between the review sections;  

The introduction includes an overview of COVID-19, its symptoms and the mechanism of infection. It also describes the relation between the coagulation system activation and the infection with COVID-19 in order to briefly introduce the subsequent parts of the review.

The review next section deals with the involvement and role of the coagulation system in different diseases and shifted gradually to its role in the viral infection (next section, hypercoagulability, and viral infections) and thereafter its association with COVID-19 and its commonly used and recent COVID-19 biomarkers.  The review includes an important relative section about the therapeutic approaches for hypercoagulability in COVID-19 including natural anticoagulants.

Because most of the currently used COVID-19 vaccines are formulated in nano form with different techniques, it was essential to review the nanotechnology involvement in COVID-19 treatment and COVID-19 vaccines loaded nanoparticles. The authors want to clarify that the activation of the coagulation system is not only a result of COVID-19 infection but also could result from vaccine administration. Therefore it was very important to discuss the relation between coagulation system activation and COVID-19 vaccines in the last section of the review.

  1. The text is relatively trivial and gives the impression of being written by researchers who do not have enough experience in medical science. The job adds absolutely nothing to the state of the knowledge of COVID-19.

Response

Concerning the experience of the authors, three authors have very good experience in the field of coagulation systems in many diseases (Please see some related published articles).

Professor/Abdel-Bakky MS

  • doi: 10.1007/s00210-020-01896-0.
  • doi: 10.1016/j.jare.2020.12.014.
  • doi: 10.1002/jbt.22287.
  • doi: 10.1016/j.lfs.2019.05.078.
  • doi: 10.1016/j.etap.2015.02.012.
  • doi: 10.1016/j.lfs.2021.119120.
  • doi: 10.4196/kjpp.2021.25.5.385.
  • doi: 10.1002/jat.2728.
  • doi: 10.3389/fphar.2018.01155. 
  • doi: 10.1007/s00204-011-0649-6.
  • doi: 10.1080/01480545.2018.1485688
  • doi: 10.1007/s00204-011-0663-8. 
  • doi: 10.1016/j.bbrc.2011.05.127. 

Dr/ Ewees MG

  • doi: 10.1080/01480545.2018.1485688.
  • doi: 10.3389/fphar.2018.01155. 
  • doi: 10.1007/s00210-019-01618-1
  • doi: 10.1016/j.jare.2020.12.014.

Dr/ Mahmoud NI

  • doi: 10.1002/jbt.22287. 
  • doi: 10.1016/j.lfs.2019.05.078.

In addition, Dr/Elham Amin and Dr/ Hamdoon A. Mohammed have a wide experience in the field of natural products isolation, structural elucidation and biological screening.  (They were responsible for the Natural products as anticoagulants part.

Also, Dr/Ahmed A. H. Abdellatif has excellent experience (more than 50 publications) in the field of Nanotechnology. (https://scholar.google.com/citations?user=kAxjfQEAAAAJ&hl=en) (https://www.scopus.com/authid/detail.uri?authorId=56602555400) 

Finally, we appreciate the effort of the third reviewer, but we do not agree with his expression “being written by researchers who do not have enough experience in medical science” and that the research sections are unrelated. We invite him to review the authors' published papers in the field related to the coagulation system involvement in different diseases, supplied DOIs).

Kind regards

Dr. Ahmed A. H. Abdellatif

Department of Pharmaceutics, College of Pharmacy, Qassim University, Qassim, 52471, Kingdom of Saudi Arabia

Mail: a.abdellatif@qu.edu.sa
